# Dynamic Morphology of the Ascending Aorta and Its Implications for Proximal Landing in Thoracic Endovascular Aortic Repair [note 1]

**DOI:** 10.3390/jcm12010070

**Published:** 2022-12-21

**Authors:** Denis Skrypnik, Marius Ante, Katrin Meisenbacher, Dorothea Kronsteiner, Matthias Hagedorn, Fabian Rengier, Florian Andre, Norbert Frey, Dittmar Böckler, Moritz S. Bischoff

**Affiliations:** 1Department of Vascular and Endovascular Surgery, University Hospital Heidelberg, 69120 Heidelberg, Germany; 2Institute of Medical Biometry and Informatics, University of Heidelberg, 69117 Heidelberg, Germany; 3Clinic for Diagnostic and Interventional Radiology, University Hospital Heidelberg, 69120 Heidelberg, Germany; 4Clinic for Cardiology, Angiology and Pneumology, University Hospital Heidelberg, 69120 Heidelberg, Germany

**Keywords:** TEVAR of ascending aorta, endograft of ascending aorta, dynamic morphology of ascending aorta, landing zone morphology, thoracic endovascular aortic repair

## Abstract

In this study, we assessed the dynamic segmental anatomy of the entire ascending aorta (AA), enabling the determination of a favorable proximal landing zone and appropriate aortic sizing for the most proximal thoracic endovascular aortic repair (TEVAR). **Methods:** Patients with a non-operated AA (diameter < 40 mm) underwent electrocardiogram-gated computed tomography angiography (ECG-CTA) of the entire AA in the systolic and diastolic phases. For each plane of each segment, the maximum and minimum diameters in the systole and diastole phases were recorded. The Wilcoxon signed-rank test was used to compare aortic size values. **Results:** A total of 100 patients were enrolled (53% male; median age 82.1 years; age range 76.8–85.1). Analysis of the dynamic plane dimensions of the AA during the cardiac cycle showed significantly higher systolic values than diastolic values (*p* < 0.001). Analysis of the proximal AA segment showed greater distal plane values than proximal plane values (*p* < 0.001), showing a reversed funnel form. At the mid-ascending segment, the dynamic values did not notably differ between the distal plane and the proximal segmental plane, demonstrating a cylindrical form. At the distal segment of the AA, the proximal plane values were larger than the distal segmental plane values (*p* < 0.001), thus generating a funnel form. **Conclusions:** The entire AA showed greater systolic than diastolic aortic dimensions throughout the cardiac cycle. The mid-ascending and distal-ascending segments showed favorable forms for TEVAR using a regular cylindrical endograft design. The most proximal segment of the AA showed a pronounced conical form; therefore, a specific endograft design should be considered.

## 1. Introduction

Thoracic endovascular aortic repair (TEVAR) with landing in the ascending aorta (AA) is a treatment option for a variety of proximal aortic pathologies in selected patients for whom open surgery carries high risk [1,2,3,4,5]. The outcome of any TEVAR procedure critically depends on the morphology of the proximal landing zone (PLZ), with non-optimal aortic sizing and endograft sizing reportedly associated with increasing rates of endoleaks (ELs), endograft migration, and reintervention [6,7,8,9,10]. The pulsatile morphology of the AA, and its variable segmental geometry during the cardiac cycle, may be disadvantageous for proximal endograft alignment and may thereby lead to poor TEVAR outcomes [11,12,13]. Few studies have reported the dynamic slice anatomy and motility of selected parts of the AA and aortic arch, and the segmental anatomy of the entire AA remains under-reported [14,15,16,17]. TEVAR in the AA shows promising outcomes but is associated with high rates of ELs and substantial rates of retrograde aortic dissection (RAD) and conversion [2,4,18,19]. Thus, the dynamic segmental anatomy of the AA must be further investigated to advance TEVAR in the AA and to improve the current clinical and technical outcomes of this procedure.

The objective of the present study was to assess the dynamic segmental anatomy of the entire AA, enabling the determination of a favorable PLZ and appropriate aortic sizing for the most proximal TEVAR.

## 2. Materials and Methods

### 2.1. Study Design and Patient Population

We conducted a single-center, retrospective analysis of prospectively collected clinical and computer tomography (CT)-based imaging data. CT examinations were clinically indicated due to the critical stenosis of the aortic valve, for the planning of transcatheter aortic valve implantation (TAVI). The retrospective scientific data analyses were approved by the local ethics committee (S-620/2018).

This study included patients with an indication for TAVI, admitted between 1 July and 8 October 2020, who underwent preoperative electrocardiogram-gated computed tomography angiography (ECG-CTA) (Philips IQon; Philips, Best, The Netherlands) of the entire AA. All the included patients had a non-operated AA. Patients were excluded if they exhibited AA pathology (e.g., aneurysm or dissection), dilatation of the AA of >40 mm, a left ventricular ejection fraction of <30%, or incomplete CTA presentation of the AA. The study did not include patients with connective tissue disease, previous surgery of the left ventricle, or with AA calcinosis of >30% of the circumference (Figure 1).

### 2.2. Image Acquisition

Image acquisition was performed using a 64-slice CT scanner (Philips IQon; Philips, Best, The Netherlands) in the supine position with an inspiratory breath-hold. We retrospectively obtained the results for ECG-gated CTA examinations of the heart, the ascending aorta, and the aortic arch with the following protocol parameters: tube potential of 120 kVp, automated tube current modulation, and 80 mL of iodinated contrast medium followed by a 50 mL saline bolus. The images were reconstructed at 5% steps of the RR interval, with a slice thickness of 0.67 mm, a slice increment of 0.33 mm, and the IMR 1, cardiac routine kernel. The reconstruction at 40% of the RR interval was defined as the diastolic phase and the reconstruction at 75% of the RR interval was considered the systolic phase.

### 2.3. Segmentation and Image Analysis

We analyzed the ECG-CTA of the entire AA in the systolic and diastolic phases. The CTA image series was uploaded from the institutional database to a separate workstation equipped with the “3mensio Vascular” postprocessing software (Pie Medical Imaging BV, Maastricht, The Netherlands). After a three-dimensional centerline reconstruction of the systolic and diastolic image series of the entire AA (from the sinotubular junction to the brachiocephalic trunk), manual aortic segmentation was performed (Figure 2). For this segmentation, a 25 mm centerline length of each AA segment was obtained, based on the recommended length of the proximal landing zone for TEVAR in the proximal thoracic aorta [2]. Each plane of each segment was automatically set perpendicular to the centerline. The proximal plane of segment A was at the sinotubular junction, the middle of the length of segment B was at the middle of the AA, and the distal border of segment C was at the proximal circumference of the brachiocephalic trunk (Figure 2). For each plane of each segment, the area and maximum and minimum diameters in the systole and diastole phases were automatically recorded (Figure 3). All image series were analyzed by two independent study collaborators.

## 3. Definitions

The centerline length of the AA was between the sinotubular junction and the proximal border of the brachiocephalic trunk. Segmental pulsatility was defined as the radial change in the aortic lumen during the cardiac cycle and was calculated as the largest difference between the systole and diastole values in both area and diameter [11]. The segmental shapes of aortic segments were determined based on the difference between the distal and proximal sizes of the aortic planes in the systole and diastole phases. Segmental strain was determined as follows: (maximum systolic diameter−maximum diastolic diameter)/maximum diastolic diameter (%) [20]. The aortic plane was defined as a 2D aortic slice positioned perpendicular to the centerline. The aortic segment was defined as a cylindrical/conical/reversed conical 3D part of the AA between the proximal and distal segmental plane.

## 4. Statistical Analysis

All the collected data were descriptively analyzed using median and Q1–Q3 for continuous variables and absolute and relative numbers for categorical variables. Intraclass correlation coefficients (ICC3), including 95% confidence intervals (CIs), were used to assess the inter-rater reliability, with the rater considered a fixed effect, since ICC2 caused unstable models. These models only represented the reliability between these raters and do not apply to other raters. The Wilcoxon signed-rank test was used to compare the aortic size values between the systole and diastole phases and between the proximal and distal aortic plane dimensions. We also calculated the non-parametric 95% CIs for the median values. An explorative significance level of *p* < 0.05 was used, but *p* values are descriptive only.

## 5. Results

This study enrolled 100 patients, including 53 males (53%), with a median age of 82.1 years (range 76.8–85.1 years), and median BMI of 25.8 kg/m^2^ (range 23.1–29.79 kg/m^2^). All the patients showed critical aortic stenosis. The leading cardiovascular risk factors were arterial hypertension (89%, 89/100), coronary heart disease (92%, 92/100), and hyperlipoproteinemia (64%, 64/100). Table 1 presents patient demographics.

### 5.1. Aortic Dimensions

The median AA length was 69.3 mm (Q1–Q3, 63.75–75.4 mm). The aortic diameter (D) and area showed considerable variation during the heart cycle (Table 2). The smallest systolic and diastolic D and area were at the proximal plane of segment A: systolic D_min_ 26.2 mm (Q1–Q3, 24.4–28.1 mm); systolic D_max_ 29.6 mm (Q1–Q3, 27.9–31.5 mm); systolic area 618 mm^2^ (Q1–Q3, 539–701 mm^2^); diastolic D_min_ 25.9 mm (Q1–Q3, 24.1–28.2 mm); diastolic D_max_ 29.2 mm (Q1–Q3, 27.4–31.1 mm); and diastolic area 614.5 mm^2^ (Q1–Q3, 516–696.5 mm^2^).

### 5.2. The 2D Form of Aortic Planes

All the cross-sections of the AA were oval, with a relative difference of approximately 10% between the maximum and minimum diameter, which was constant throughout the cardiac cycle (Table 3). Analysis of the dynamic plane dimensions of the AA during the cardiac cycle revealed that systolic values were significantly higher than diastolic values (*p* < 0.001) (Table 4). The systolic D of segment A was 0.3 mm larger than the diastolic D on the proximal plane (95% CI 0.15–0.55) and the distal plane (95% CI 0.25–0.55). Similarly, on the proximal and distal planes of segment B, the systolic D was 0.5 mm (95% CI 0.35–0.6) larger than the diastolic D. At segment C, the dimension variability during the cardiac cycle was more pronounced on the distal segmental plane (0.5 mm, 95% CI 0.4–0.75) than on the proximal segmental plane (0.3 mm, 95% CI 0.2–0.5). Similar to the plane diameter, the segmental area was larger in the systole phase than in the diastole phase (*p* < 0.001) (Table 4).

### 5.3. The 3D Form of Aortic Segments

The 3D segmental form was described using a comparison between the distal and proximal plane sizes of each AA segment, considering that all the segments lied on the centerline and had a fixed length (25 mm). Analysis of segment A revealed larger distal plane values than proximal plane values (*p* < 0.001) (Table 5). Therefore, segment A of the AA had a reversed funnel form. In segment B, the dynamic values did not notably differ between the distal plane and the proximal segmental plane, thus resulting in a cylindrical form for segment B of the AA during the cardiac cycle (Table 5). In segment C, during the cardiac cycle, the proximal plane values were larger than the distal segmental plane values (*p* < 0.001); therefore, segment C had a funnel form (Table 5).

The aortic strain was above 5% in all the planes of the AA (Table 6). The greatest variation was observed in the strain at the distal plane of segment C (1.8 ± 2.9%) on the border of the aortic arch.

Image quality was sufficient for reliable measurements in all the cases (*N* = 100). The intraclass correlation coefficient was >0.91, indicating high similarity between the measured values and good reproducibility for all the measurements.

## 6. Discussion

The current study shows a predominance of the systolic over diastolic diameter during the whole cardiac cycle at all levels of the ascending aorta. Each aortic plane demonstrated an oval-shaped 2D morphology with a 10% predominance of maximum plane diameters over small diameters. Furthermore, our analysis revealed a cylindrical form for the mid-ascending aortic segment, a slightly funneled form for the distal-ascending segments, and a pronounced conical form for most proximal segments of the AA.

The currently available reports in the literature highlight the pulsatility of some segments of the AA. De Heer et al. found that the aortic diameter at the sinotubular junction is larger in the systole phase than in the diastole phase (D_max_ systolic 32.4 ± 3.8 mm, D_max_ diastolic 31.5 ± 3.9 mm, *p* < 0.001) [17]. Jian-ping et al. reported significant changes in the aortic diameter of the distal AA during the cardiac cycle, with greater aortic size in the systole phase than in the diastole phase (3.26 ± 0.24 mm and 3.18 ± 0.27 mm, respectively, *p* < 0.01) [13]. Rengier et al. showed the prominent mid-ascending pulsatility of the AA in healthy volunteers, where the systolic aortic dimension was over 10% greater than the diastolic aortic dimension [14]. In line with these prior reports, our current findings showed a wide range of variability in the cross-sectional dimensions of the aorta during the cardiac cycle, with clearly larger systolic dimensions than diastolic dimensions at all levels of the AA (*p* ≤ 0.001).

Satriano et al. performed a 3D reconstruction of the ECG-CTA series and reported asymmetrical distension in the AA during the cardiac cycle, which was more prominent along the greater curvature of the AA, consistent with the jet flow direction during heart output [21]. Other reports have described non-circular shapes for some aortic planes during the cardiac cycle [11,13,22].

Liu et al. analyzed the precise sizing for TEVAR in the AA and reported that if the diameters differed by >5%, the real aortic diameter should be calculated as an average between the maximum and minimum diameters, to avoid retrograde aortic dissection [22]. Our present findings confirmed the oval shape of the segmental planes at all levels of the AA, with a relative difference of approximately 10% between the maximum and minimum diameter throughout the cardiac cycle. Thus, it seems appropriate to use the average diameter for the precise measurement of the AA diameter.

In our current study, we observed increased aortic diameter in the systole phase compared with that in the diastole phase and showed an AA strain of up to 5%. Satriano et al. reported a 10.2 ± 6.0% peak principal strain amplitude for the entire AA [21]. Redheuil et al. reported a similar AA strain (8 ± 4%) in patients over 70 years old and found an AA strain of up to 15 ± 8% in patients 40–49 years old [23]. Thus, the published literature and the data from our current study support the use of a systolic CTA series for the most precise sizing of TEVAR in the AA, wherein 5–15% of the aortic diameter size may be balanced out compared with CTA in the diastolic phase, independent of the patient’s age.

In a recent systematic review, Muetterties et al. reported an 18.6% rate of early-term EL Ia after TEVAR in the AA [4]. Similarly, a meta-analysis by Baikoussis et al. revealed a high pooled rate of late EL Ia (16.4%) after TEVAR in the AA [2]. These results are most likely related to the inappropriate alignment of the endograft to the aortic wall [4]. Accordingly, it is crucial to understand the 3D shape of the PLZ to improve outcomes.

A study by van Prehn et al. reported the dynamic plane morphology at the three AA levels and described the 3D motions of 2D aortic planes. However, the authors did not consider the 3D segmental morphology of AA, which is essential for understanding the volume geometry of a potential proximal landing zone [24].

In the current study, we observed that the mid-ascending segment of the AA retained its cylindrical shape throughout the cardiac cycle; therefore, the common cylindrical design of the endograft seems appropriate in this setting. The distal AA segment showed a funnel form; however, the diameter size difference of 1.5 mm between the proximal and distal segmental planes does not seem to be relevant for practical sizing; therefore, a cylindrical endograft design could also be considered here. In contrast, most proximal AA segments showed a reversed funnel (conical) form, which is reportedly unfavorable for aortic endograft alignment [25,26,27]. Moreover, the difference of >5 mm between the proximal (smaller) and distal (larger) diameters of segmental planes corresponds to an 18% (5.5/29.6 mm) systolic diameter difference between the proximal and distal segmental planes. Therefore, the conventional cylindrical endograft design may not be suitable for use in such cases.

## 7. Limitations

The present study has several limitations. First, we included patients with significant aortic stenosis, which may influence aortic asymmetry throughout the power and direction of jets during the cardiac cycle. However, previously published studies reported increased arterial stiffness for the whole arterial tree, including the AA, due to severe aortic stenosis; however, a reduced distensibility (a function of change in AA diameter and arterial pressure) of non-calcified AA was not observed compared with patients without several aortic stenoses if cardiac output and stroke volume were saved [28,29]. Furthermore, evenly distributed AA stiffness may introduce bias in terms of absolute diameter and area numbers. However, this is unlikely to result in any change in the aortic plane size ratios. Thus, the volumetric form of the AA segments would probably stay the same.

Second, our patient cohort included those with advanced age and atherosclerosis, which may influence aortic distensibility. One may speculate that AA compliance may be higher in younger subjects. Third, this study did not investigate the longitudinal motions, side deviations, or angulation of the AA during the cardiac cycle, which may be relevant for a complete description of 3D aortic geometry during the cardiac cycle.

## 8. Conclusions

The entire AA showed variable dynamic anatomy during the cardiac cycle. Precise AA sizing, using the average aortic diameter and systolic CTA series, may be considered. The mid-ascending and distal-ascending segments showed favorable forms for TEVAR using a regular cylindrical endograft design. The most proximal segment of the AA showed a pronounced conical form; therefore, a specific endograft design should be considered.

## Figures and Tables

**Figure 1 jcm-12-00070-f001:**
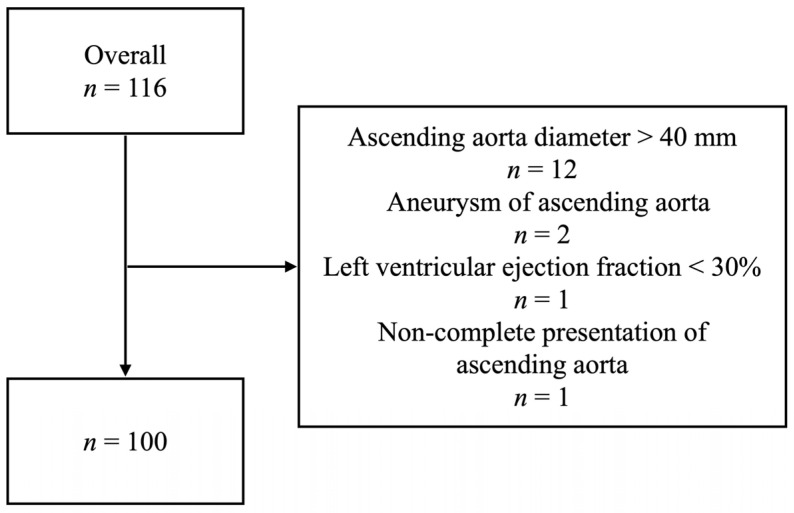
Flowchart of patient inclusion. The chart demonstrates selection of patients in terms of inclusion/exclusion criteria.

**Figure 2 jcm-12-00070-f002:**
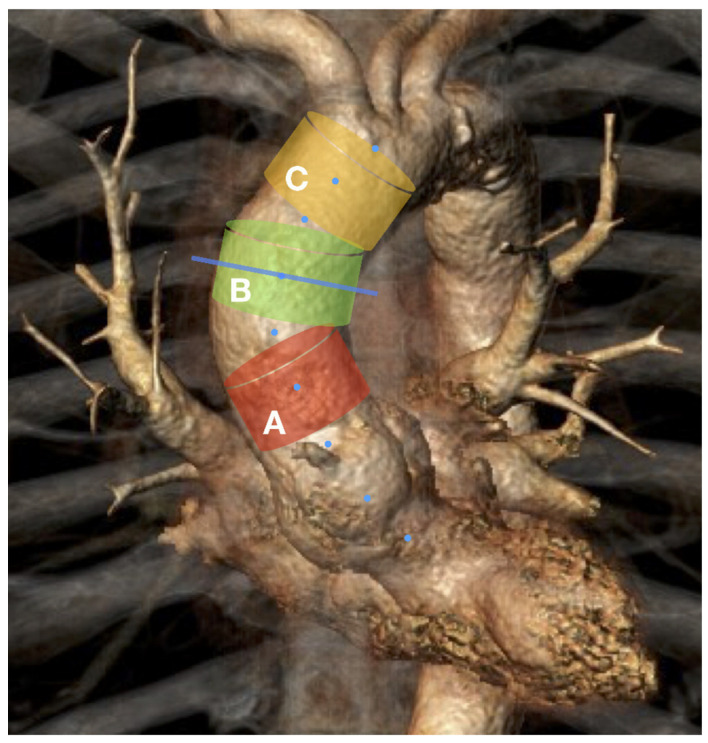
Segmentation of the ascending aorta: Segment A: 2.5 cm distally to the sinotubular junction. Segment B: middle of the segment is on the middle line of the ascending aorta. Segment C: 2.5 cm proximally to the brachiocephalic trunk. All segments had a 25 mm centerline length. Dashed blue line is the centerline. Solid transverse blue line is the middle of the ascending aorta.

**Figure 3 jcm-12-00070-f003:**
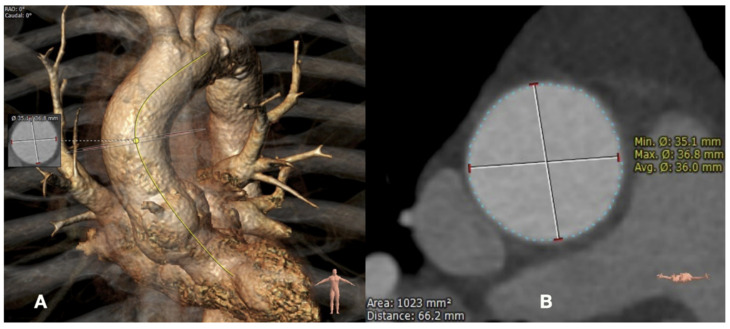
Screenshot of the “3mensio Vascular” program: (**A**) a 3D reconstruction of the ascending aorta. Solid yellow line is the centerline; (**B**) axial presentation of the mid-ascending plane of the ascending aorta with automatically measured size values.

**Table 1 jcm-12-00070-t001:** Patient demographic data ^a^.

Variable	Median [Q1–Q3]; % (*n*/*N*)
Age, years	82.1 [76.8–85.1]
Male	53 (53/100)
Female	47 (47/100)
BMI, kg/m^2^	25.8 [23.1–29.7]
Hypertension	89 (89/100)
Coronary heart disease	92 (92/100)
PTCA	34 (34/100)
AF	39 (39/100)
Previous stroke/TIA	14 (14/100)
COPD	13 (13/100)
Diabetes	28 (28/100)
Adipositas (BMI > 30)	22 (22/100)
Chronic renal insufficiency (Creatinine > 1.2 mg/dL)	18 (18/100)
History of smoking	29 (29/100)
Hyperlipoproteinemia	64 (64/100)
Stenosis of aortic valve	100 (100/100)

Abbreviations: BMI, body mass index; PTCA, percutaneous transluminal coronary angioplasty; AF, atrial fibrillation; TIA, transient ischemic attack; COPD, chronic obstructive pulmonary disease. ^a^ Categorical data are presented as absolute numbers and percentages; continuous data are presented as medians and interquartile ranges (*N* = 100).

**Table 2 jcm-12-00070-t002:** Segmental area and systolic/diastolic diameter values ^a^.

Segment	A		B		C	
Plane	Proximal Plane	Distal Plane	Proximal Plane	Distal Plane	Proximal Plane	Distal Plane
Systolic D_min_, mm	26.2 (24.4–28.1)	32.3 (30–34.7)	31.4 (29.6–34.5)	32.2 (29.9–34.1)	32.3 (30.2–34)	30.4 (28.3–32.2)
Systolic D_max_, mm	29.6 (27.9–31.5)	35.1 (32.75–37)	34.3 (32–36.75)	34.7 (32.5–36.5)	34.8 (32.7–36.5)	33.5 (31.4–35.5)
Area systolic, mm^2^	618 (539–701)	890 (777–1016)	847.5 (755.4–997)	876.5 (765–992)	883 (783–980.3)	803 (697.5–900)
Diastolic D_min_, mm	25.9 (24.1–28.2)	32.1 (30–34.1)	31.3 (29.1–34)	31.8 (29.25–34)	32.2 (30–34)	30.3 (27.9–31.6)
Diastolic D_max_, mm	29.2 (27.4–31.1)	34.9 (32.2–36.4)	33.8 (31.75–36.3)	34.3 (31.5–36)	34.5 (32–36.2)	33.2 (30.75–35.2)
Area diastolic, mm^2^	614.5 (516–696.5)	875.5 (753–986)	833.5 (732.5–974)	862.5 (740.5–978)	875 (757.5–965)	783 (679–874.5)

^a^ Data are presented as medians and quantiles (Q1, Q3) (*N* = 100).

**Table 3 jcm-12-00070-t003:** The predominance of larger over smaller diameters of the aortic planes ^a^.

Segment	Proximal Segmental Plane	Distal Segmental Plane	Proximal Segmental Plane	Distal Segmental Plane
	Systole		Diastole	
A	0.1 (0.1, 0.1)	0.1 (0.1, 0.1)	0.1 (0.1, 0.1)	0.1 (0, 0.1)
B	0.1 (0.1, 0.1)	0.1 (0.1, 0.1)	0.1 (0, 0.1)	0.1 (0.1, 0.1)
C	0.1 (0.1, 0.1)	0.1 (0.1, 0.1)	0.1 (0, 0.1)	0.1 (0.1, 0.1)

The table shows that maximum diameter of each aortic plane was larger than a smaller diameter for 10% during the cardiac cycle, thus demonstrating an oval-shaped rather than round 2D morphology. ^a^ The predominance of larger over smaller diameter was calculated for each plane as (D_max_−D_min_)/D_max_ in the systole and diastole phases; data are presented as medians and quantiles (Q1, Q3) (*N* = 100).

**Table 4 jcm-12-00070-t004:** Difference between maximum systolic and maximum diastolic diameters and areas ^a^.

Segment	Proximal Segmental Plane, mm	*p*-Value	Proximal Segmental Plane, mm^2^	*p*-Value	Distal Segmental Plane, mm	*p*-Value	Distal Segmental Plane, mm^2^	*p*-Value
A	0.3 (0.15, 0.55)	0.001	10 (5.5, 19)	<0.001	0.3 (0.25, 0.55)	<0.001	15 (13, 23)	<0.001
B	0.5 (0.35, 0.6)	<0.001	18 (14, 25.5)	<0.001	0.5 (0.35, 0.6)	<0.001	18 (16.5, 25.5)	<0.001
C	0.3 (0.2, 0.5)	<0.001	15 (11, 20)	<0.001	0.5 (0.4, 0.75)	<0.001	19 (15.5, 24)	<0.001

^a^ Differences between maximum systolic and diastolic diameters and areas are presented as medians and 95% CIs (*N* = 100).

**Table 5 jcm-12-00070-t005:** Segmental shapes of ascending aorta during the heart cycle ^a^.

Segment	D Systolic Max, mm	*p*-Value	Area Systolic, mm^2^	*p*-Value	D Diastolic Max, mm	*p*-Value	Area Diastolic, mm^2^	*p*-Value
A	5.5 (4.9, 5.8)	<0.001	293 (259, 306)	<0.001	5.3 (4.8, 5.7)	<0.001	265 (252, 293)	<0.001
B	0 (−0.3, 0.35)	0.947	1.7 (−13, 21) ^b^	0.653	0 (−0.35, 0.35)	0.99	5 (−16, 16)	0.99
C	−1 (−1.35, −0.85)	<0.001	−78 (−87.4, −59.5)	<0.001	−1.4 (−1.6, 1)	<0.001	−82 (−97, −70)	<0.001

^a^ Differences between distal and proximal maximum plane diameters and areas are presented as medians and 95% CIs (*N* = 100). ^b^ Negative values show that distal diameters and areas are smaller than proximal diameters and areas.

**Table 6 jcm-12-00070-t006:** Segmental strain of ascending aorta during the cardiac cycle ^a^.

Segment	Proximal Segmental Plane, %	Distal Segmental Plane, %
A	1.2 ± 3.7	1.2 ± 2.6
B	1.4 ± 2.5	1.4 ± 2.0
C	1.0 ± 2.3	1.8 ± 2.9

^a^ Strain was calculated as (D_max_ systolic−D_max_ diastolic)/D_max_ diastolic (%) and is presented as mean ± SD (*N* = 100).

## Data Availability

Not applicable.

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
