# Peer review of "Dynamic Morphology of the Ascending Aorta and Its Implications for Proximal Landing in Thoracic Endovascular Aortic Repairâ€"

_jcm, 2022, doi:10.3390/jcm12010070_

Round 1
Reviewer 1 Report
To author
Thank you for reporting about ‘Dynamic Morphology of the Ascending Aorta and its Implications for Proximal Landing in Thoracic Endovascular Aortic Repair’
This paper also made us think about whether the ascending aorta is suitable for landing during in TEVAR and its impact on sizing and endoleak.
I have a few little concerns, below.
1. The diameter changes in ascending aortic during the cardiac cycle and the changes in aortic diameter by site were interesting. Is there any change in aortic diameter distal the ascending aorta?
2. The data in Table 3 looks almost identical and is difficult to understand. Please make it a little clearer.
3. In aortic stenosis, accelerated blood flow is observed at the aortic valve. Does this affect the ascending aorta?
Author Response
Reviewer’s question 1: The diameter changes in ascending aortic during the
cardiac cycle and the changes in aortic diameter by site were interesting. Is there
any change in aortic diameter distal the ascending aorta?
Author’s response 1: We are grateful for your question. The diameter changes
of the aortic arch during the cardiac cycle were not investigated in this study. However, some papers reported certain changes: Dynamic Cine-CT Angiography for
the Evaluation of the Thoracic Aorta Insight in Dynamic Changes with Implications
for Thoracic Endograft Treatment, B.E. Muhs et al.; Dynamic Changes in the Aorta
During the Cardiac Cycle Analyzed by ECG-Gated Computed Tomography
Wenying Zhu et al.
Reviewer’s question 2: The data in Table 3 looks almost identical and is difficult to
understand. Please make it a little clearer.
Author’s response 2: Thank you for this comment. The table and the
comments were modified to improve readability (page 6, rows 187-191).
Reviewer’s question 3: In aortic stenosis, accelerated blood flow is observed at the
aortic valve. Does this affect the ascending aorta?
Author’ response 3: This is an interesting comment and we agree, that
potentially, the high-grade aortic valve stenosis may introduce some bias regarding
absolute diameter and area numbers (limitation section page 9 rows 282-291).
However, it is unlikely that this would result in changes of the aortic plane size ratios.
Moreover, the formula to calculate aortic distensibility is known [“Aortic distensibility was calculated using the formula: 2 x (change in aortic diameter)/(diastolic aortic
diameter) x (change in aortic pressure), where change in aortic diameter = systolic
minus diastolic aortic diameter, change in aortic pressure = systolic minus diastolic
aortic pressure]“. (Distensibility of the ascending aorta: comparison of invasive and
non-invasive techniques in healthy men and in men with coronary artery disease
- STEFANADIS et al.; Greenfield JC Jr, Patel DJ. Relation between pressure and
diameter in the ascending aorta of man.; Caro CG, Pedley TJ, Schroten RC, Seed
- The Mechanics of the Circulation. Oxford: Oxford University Press, 197).
Which doesn’t include the grade of aortic valve stenosis or velocity of the blood jet coming into ascending aorta.

Reviewer 2 Report
Consider examining whether there was any relationship between the severity of aortic valve stenosis and the degree of segmental pulsatility or strain.
Table 3 is confusing, and the authors should consider include a larger number of significant figures.
It would be helpful to include a diagram that demonstrated the average funnel shapes for segments A (reverse funnel), B (cylindrical) and C (funnel).
Consider reordering the discussion so that the main findings are presented in the opening paragraph of the discussion.
Author Response
Reviewer’s question 1: Consider examining whether there was any relationship
between the severity of aortic valve stenosis and the degree of segmental pulsatility
or strain.
Author’s response 1: We are grateful for your question and we are agree, that
potentially, the high-grade aortic valve stenosis may introduce some bias regarding
absolute diameter and area numbers (limitation section page 9 rows 282-291).
However, it is unlikely that this would result in changes of the aortic plane size ratios.
Moreover, the formula to calculate aortic distensibility is known [“Aortic distensibility was calculated using the formula: 2 x (change in aortic diameter)/(diastolic aortic
diameter) x (change in aortic pressure), where change in aortic diameter = systolic
minus diastolic aortic diameter, change in aortic pressure = systolic minus diastolic
aortic pressure]“. (Distensibility of the ascending aorta: comparison of invasive and
non-invasive techniques in healthy men and in men with coronary artery disease
- STEFANADIS et al.; Greenfield JC Jr, Patel DJ. Relation between pressure and
diameter in the ascending aorta of man.; Caro CG, Pedley TJ, Schroten RC, Seed
- The Mechanics of the Circulation. Oxford: Oxford University Press, 197).
Which doesn’t include the grade of aortic valve stenosis or velocity of the blood jet coming into ascending aorta.
Reviewer’s comment 2: Table 3 is confusing, and the authors should consider include
a larger number of significant figures.
Author’s response 1: We appreciate your question. The table and the
comments were modified to improve the understanding (page 6, rows 187-191).
Reviewer’s comment 3: It would be helpful to include a diagram that demonstrated the average funnel shapes for segments A (reverse funnel), B (cylindrical) and C (funnel).
Author’s response 3: Thank you for this comment. We agree, that
graphical presentation matters, that’s why we’ve constructed a graphical abstract,
which visually presented the main findings of our study (see attachment file).
Reviewer’s comment 4: Consider reordering the discussion so that the main
findings are presented in the opening paragraph of the discussion.
Author’ response 4: we are very grateful for your comment. The discussion session
was modified (page 8, rows 222-227).
The corrections were red marked.
The proof reading was made by JCM editing service.
